# Differences in Changes in Game Usage Time and Game Use-Related Factors Depending on Parity in a Prospective Study of Pregnant Women in Japan

**DOI:** 10.3390/healthcare11233017

**Published:** 2023-11-22

**Authors:** Hiroko Sato, Toshiyuki Yasui

**Affiliations:** 1Graduate School of Health Sciences, Tokushima University, Tokushima 770-8503, Japan; 2Department of Midwifery, Tokushima University Graduate School, Tokushima 770-8503, Japan; 3Department of Reproductive and Menopausal Medicine, Tokushima University Graduate School, Tokushima 770-8503, Japan; tosyasui@tokushima-u.ac.jp

**Keywords:** pregnant woman, game usage, gaming disorder, prospective study

## Abstract

Game usage has recently been increasing, but the actual situation of game usage and issues among pregnant women are not clarified. The purpose of this prospective longitudinal study was to examine changes in game usage, lifestyle, and thoughts about game usage during pregnancy depending on parity and to clarify the characteristics of pregnant women who continue to use games. We conducted three web surveys in early, mid- and late pregnancy in 238 pregnant women. For primiparous women who continued to use games, there was a significant increase in game usage time from early to late pregnancy (*p* = 0.022), and 25.0% of those women had anxiety that they might have a game addiction. For primiparous women in mid-pregnancy and multiparous women in early and late pregnancy, the proportions of women who thought that they could not use gaming sufficiently due to pregnancy and child-rearing were significantly higher in women who continued to use games. In both primiparous women and multiparous women, the proportion of partners who used games was significantly higher in women who continued to use games. It is necessary for midwives to discuss with pregnant women and their partners about game usage and to provide advice about control of game usage in daily life.

## 1. Introduction

The population engaged in game usage has been increasing globally [1], and a subset of this population has problematic game usage [2]. It was reported that the frequency of video game use was associated with self-worth and social acceptance for young women negatively [3]. In addition, Stockdale et al. reported that excessive gaming was related to depressive symptoms and decreased feelings of parental efficacy among mothers [4]. Nakayama et al. reported that problematic game usage was related to the young age of the people with habitual game usage [5]. Since game usage has recently been increasing in the younger generation, the number of people with problematic game usage may increase in the future. There have been few reports on the actual situation of game usage and issues regarding game usage among pregnant women. We previously reported that the proportion of pregnant women using games is approximately 40% in early pregnancy [6] and that women tend to stop playing games or reduce their game usage time after pregnancy [7]. However, changes in game usage and game usage time with the advance of the gestational period have not been reported. Considering that there are differences in daily life such as the time spent on childcare or sleep [8] and the status of nutritional intake [9] between primiparous women and multiparous women, there may be different characteristics of game usage according to parity.

We previously reported that although some women stop playing games when they become pregnant, there are some women who continue to play games after pregnancy. Moreover, women who continue to play games after pregnancy are likely to have anxiety that they might have a game addiction, and it is likely that their partners also play games [7]. We also reported that there was a correlation between game usage time and anxiety in women in early pregnancy about whether they might have a game addiction [6]. However, changes in the characteristics of game usage in pregnant women who continue to use games during their subsequent pregnancy period have not been clarified. Various physical and psychological changes occur during pregnancy. It has been reported that hormonal fluctuations and anatomical changes during pregnancy induce sleep deficiency [10,11] and that the proportion of pregnant women engaging in physical activity decreases with the advance of the gestational period [12,13]. Anxiety about giving birth [14] and a desire for improvement in lifestyle have also been reported for pregnant women since pregnant women become more concerned about the health of the mother and fetus [15]. Thoughts about game usage may also change in pregnant women with the advance of the gestational period. The purpose of this prospective longitudinal study was to examine changes in game usage, lifestyle, and thoughts about game usage during pregnancy in primiparous women and multiparous women and to clarify the characteristics of pregnant women who continue to use games.

## 2. Materials and Methods

### 2.1. Design

We conducted this prospective study between April 2021 and December 2022 at a birth center for low-risk pregnant women. This birth center handles approximately 700 births annually in one provincial city in Japan. The necessary sample size was determined to be 172 by using effect size (0.3), α coefficient (0.05), power (0.95), and degree of freedom (2). The sample size was determined to be 645 considering the number of uncollected samples.

### 2.2. Participants

A total of 693 pregnant women who visited the birth center for a medical check during their first trimester were recruited and were informed about the study’s objectives and methodologies. Of those women, 645 pregnant women agreed to take part in the study. A flow diagram of this prospective study is shown in Figure 1. QR codes for the online survey were distributed to the participants, and the survey was conducted by using Survey Monkey, an online tool for creating and managing questionnaires. The participants were informed that they were deemed to have consented to participate in the study by checking a box for consent “agree” before they started answering an online questionnaire. With regard to consent for participation from each subject, we explained that each subject could participate in the study voluntarily. It was clarified that refusing to participate would not cause harm, and this assurance was provided to all potential participants. Additionally, participants were informed that the collected data would exclusively serve the purpose of the study but not for any other objectives. The survey conducted online ensured the participants’ anonymity. Response rates to the questionnaires were 62.5% (403/645) at early pregnancy, 47.9% (309/645) at mid-pregnancy and 37.1% (239/645) at late pregnancy. We analyzed data for 238 women (238/645: 36.9%) for whom there were responses at all three time points.

### 2.3. Data Collection

Women over 20 years of age who were married (or have a partner) and who intended to give birth at this birth center were included. Women with multiple pregnancies and women facing challenges in completing the questionnaire due to mental or physical issues were excluded. We also excluded women such as those with a previous history of severe hypertension, diabetes, schizophrenia, or severe depression before pregnancy and those currently undergoing treatment for these conditions. We recruited subjects who met the inclusion and exclusion criteria and provided an explanation sheet of the study.

The questionnaire for early pregnancy included four sections. The first section was about background characteristics; the second section was about game usage during pregnancy; the third section was about daily life behavior and thoughts about game usage; and the fourth section was about the Internet Gaming Disorder Scale (IGDS), which consists of nine dichotomous items related to Internet Gaming Disorder (IGD). In the first section, we asked about employment (working, not working), smoking habits (never, used to smoke but stopped before pregnancy, used to smoke but stopped after pregnancy, current smoking), alcohol consumption habits (never, used to drink but stopped before pregnancy, used to drink but stopped after pregnancy, current drinking), partner’s age, and the partner’s game usage in the past month. The second section comprised inquiries concerning game use and the daily time allocated to playing games in the past month. In the third section, we asked about the following: average daily sleeping hours; consistency in wake-up time and bedtime; occurrence of days when insufficient sleep was due to game usage; occurrence of days when meals were not cooked due to game usage; occurrence of days without regular meals (3 meals/day) due to game usage; frequency of consuming ready-to-eat meals (e.g., instant food, precooked food, and fast food); perception of game usage negatively affecting a child’s or children’s development; perception of game usage being addictive; experience of ever feeling addicted to games; and feelings of not having enough time to play games due to pregnancy or child-rearing. In the fourth section, we used the IGDS, which consists of nine dichotomous items related to IGD, where each response is either ‘yes’ (1 point) or ‘no’ (0 points), with a cutoff value set at 5 points [16]. Sumi et al. translated the IGDS into Japanese [17]. The questionnaire for mid-pregnancy also consisted of four sections. In the first section, only two questions on gestational weeks of the pregnant woman and the partner’s game usage were included. The questions in the second, third, and fourth sections were the same as those at early pregnancy. The questionnaire in late pregnancy was exactly the same as that in mid-pregnancy. Information including age and parity from medical records was obtained about each subject with consent.

### 2.4. Data Analysis

Descriptive statistics were used to analyze the background characteristics of the subjects. The Shapiro–Wilk test was used to examine the normality of the variables. Comparisons of sleeping hours for women in the three periods and comparisons of game usage times for women in the three periods were performed by Friedman’s test with a Bonferroni correction for post hoc analysis. The Mann–Whitney U test was used to compare age, partner’s age, sleeping hours, and game usage time between the two groups in parity. In the analysis, the gaming usage time of the pregnant women who did not play games was considered as 0 min. The chi-square test or Fisher’s exact test with a Bonferroni correction was used for comparisons between the two groups in parity, employment, smoking habit, alcohol drinking habit before pregnancy, game usage, partner’s game usage, daily life behaviors, thoughts on games, and the IGDS score.

In order to compare game usage time and various factors that may be related to game usage, the women were divided into three groups: Group A (women who played games in all three periods), Group B (women for whom there was a change in game usage in the three periods), and Group C (women who did not play games in any of the three periods). Comparisons of game usage times of women in Group A in the three periods and comparisons of game usage times of women in Group B in the three periods were performed by Friedman’s test with a Bonferroni correction for post hoc analysis. In the analysis of Group B, the gaming usage time of the pregnant women who did not play games was considered as 0 min. Comparisons of sleeping hours for women, and age of the partner in the three groups (Group A, Group B, and Group C) were performed by the Kruskal–Wallis test with a Bonferroni correction for post hoc analysis. The chi-square test or Fisher’s exact test with a Bonferroni correction was used for comparisons of employment, smoking habit, alcohol drinking habit before pregnancy, game usage, partner’s game usage, daily life behaviors, and thoughts on games among the three groups. We set a *p* value of less than 0.05 as a statistical significance. We conducted all statistical analyses by using SPSS statistics ver.28.0 (IBM Corp., Armonk, NY, USA).

## 3. Results

### 3.1. Background Characteristics of All Subjects

The mean numbers (±standard deviation: SD) of gestational weeks at the time of the questionnaire response were 10.8 (±1.9) at early pregnancy, 26.2 (±1.1) at mid-pregnancy, and 36.4 (±1.1) at late pregnancy. The mean (±SD) age of pregnant women was 31.6 (±4.7) years. Out of 238 pregnant women, 190 women (79.8%) were working. In terms of smoking, 183 women (76.9%) had never smoked, 39 women (16.4%) used to smoke but stopped before pregnancy, 14 women (5.9%) used to smoke but stopped after pregnancy, and two women (0.8%) were smoking. In terms of alcohol consumption, 43 women (18.1%) had never drunk alcohol, 122 women (51.3%) used to drink but stopped before pregnancy, and 73 women (30.7%) used to drink but stopped after pregnancy. The mean (±SD) durations of sleeping time per day in the 238 women were 7.4 (±1.4) h in early pregnancy, 7.0 (±1.1) h in mid-pregnancy, and 6.7 (±1.4) h in late pregnancy. Sleeping time in mid-pregnancy and that in late pregnancy were both significantly shorter than that in early pregnancy (*p* = 0.003 and *p* < 0.001, respectively). There was no significant difference in the proportions of women who played games in the three periods (42.4% in early pregnancy, 33.6% in mid-pregnancy, and 34.9% in late pregnancy).

### 3.2. Comparison of Background Characteristics and Various Factors in Primiparous Women and Multiparous Women

The age of multiparous women (32.6 ± 4.5 years) was significantly greater than that of primiparous women (30.5 ± 4.6 years) (*p* < 0.001). There were no significant differences in background characteristics such as age of the partner, employment, smoking habit, and alcohol drinking before pregnancy between primiparous women and multiparous women. The median periods of game usage time per day (10–90 percentile) for primiparous women were 0.0 (0.0–120.0) min in early pregnancy, 0.0 (0.0–120.0) min in mid-pregnancy, and 0.0 (0.0–180.0) min in late pregnancy. The median periods of game usage time per day (10–90 percentile) for multiparous women were 0.0 (0.0–60.0) min in early pregnancy, 0.0 (0.0–60.0) min in mid-pregnancy, and 0.0 (0.0–60.0) min in late pregnancy. There were no significant differences in game usage time among the three periods for both primiparous women and multiparous women. In early pregnancy, the game usage time in primiparous women was significantly longer than that in multiparous women (*p* = 0.006).

A comparison of background characteristics and various factors for primiparous women (*n* = 113, 47.5%) and multiparous women (*n* = 125, 52.5%) is shown in Table 1. We divided responses to daily life behaviors and thoughts on games into “yes” (frequently and sometimes) and “no” (not at all) for comparison. In both primiparous women and multiparous women, the sleeping hours in late pregnancy were significantly shorter than those in early pregnancy (*p* = 0.007 and *p* < 0.001, respectively). In all three periods, the sleeping hours in multiparous women were significantly shorter than those in primiparous women (*p* = 0.008, *p* = 0.007, and *p* = 0.014, respectively).

In multiparous women, there were significant differences in the proportion of women with regularity of wake-up time and bedtime and the proportion of women with a high frequency of eating ready-to-eat meals among the three pregnancy periods. In early pregnancy, the proportion of primiparous women who reported that there were days on which they could not cook their own meals due to game use was higher than the proportion of multiparous women (*p* = 0.049).

### 3.3. Comparison of Background Characteristics and Various Factors among the Three Groups According to Changes in Game Usage during Pregnancy

We divided the 238 women based on changes in their game usage during pregnancy into three groups: Group A (*n* = 59, 24.8%), Group B (*n* = 63, 26.5%), and Group C (*n* = 116, 48.7%). There were no significant differences in the proportions of primiparous women and multiparous women among the three groups (Group A: 54.2% and 45.8%, respectively; Group B: 49.2% and 50.8%, respectively; Group C: 43.1% and 56.9%, respectively). We compared background characteristics of the 238 women among Group A, Group B, and Group C and compared those between primiparous women and multiparous women as well. There were no significant differences in the background characteristics (age of partner, employment, smoking habit, alcohol drinking before pregnancy) among Group A, Group B, and Group C. There were also no significant differences in these background characteristics between primiparous women and multiparous women.

The median periods of game usage time per day (10–90 percentile) for women in Group A were 60.0 (20.0–180.0) min in early pregnancy, 60.0 (20.0–180.0) min in mid-pregnancy, and 90.0 (30.0–180.0) min in late pregnancy, and there was a significant increase from early pregnancy to late pregnancy (*p* = 0.013). Changes in game usage time of pregnant women in Group A for primiparous women and multiparous women are shown in Figure 2.

For primiparous women in Group A, the periods of game usage were 60.0 (10.0–300.0) min in early pregnancy, 60.0 (30.0–210.0) min in mid-pregnancy, and 120.0 (20.0–360.0) min in late pregnancy, and there was a significant increase from early pregnancy to late pregnancy (*p* = 0.022). On the other hand, for multiparous women in Group A, the periods of game usage were 60.0 (20.0–132.0) min in early pregnancy, 60.0 (10.0–180.0) min in mid-pregnancy, and 60.0 (20.0–186.0) min in late pregnancy, and there were no significant differences among the three periods.

The median periods of game usage time (10–90 percentile) per day for women in Group B were 30.0 (0.0–120.0) min in early pregnancy, 0.0 (0.0–66.0) min in mid-pregnancy, and 0.0 (0.0–96.0) min in late pregnancy, and there was a significant decrease from early pregnancy to mid-pregnancy (*p* = 0.033). For primiparous women in Group B, the periods of game usage were 30.0 (0.0–168.0) min in early pregnancy, 0.0 (0.0–84.0) min in mid-pregnancy, and 0.0 (0.0–120.0) min in late pregnancy, and there were significant decreases from early pregnancy to mid-pregnancy and late pregnancy (*p* = 0.047 and *p* = 0.033, respectively). For multiparous women in Group B, the periods of game usage were 10.0 (0.0–60.0) min in early pregnancy, 0.0 (0.0–67.0) min in mid-pregnancy, and 0.0 (0.0–60.0) min in late pregnancy, and there was no significant difference in game usage time among the three periods. In early pregnancy and late pregnancy, game usage times in primiparous women were significantly longer than those in multiparous women in both Group A and Group B (Group A: *p* = 0.021 and *p* = 0.004, respectively; Group B: *p* = 0.01 and *p* = 0.01, respectively). In all three pregnancy periods, game usage time of women in Group B were significantly shorter than women in Group A in both primiparous women and multiparous women (early pregnancy: *p* = 0.018 and *p* < 0.001, respectively; mid-pregnancy: *p* < 0.001 and *p* < 0.001, respectively; early pregnancy: *p* < 0.001 and *p* < 0.001, respectively).

In all three groups, mean sleeping hours in late pregnancy were shorter than those in early pregnancy (*p* = 0.005, *p* = 0.001, and *p* < 0.001, respectively), and mean sleeping hours of multiparous women in Group A and Group C were shorter in late pregnancy than in early pregnancy (*p* = 0.016 and *p* = 0.005, respectively). In Group A, the proportion of women who consumed ready-to-eat meals at a frequency of three or more days per week was higher in early pregnancy than in mid-pregnancy and late pregnancy (*p* = 0.006), and a significant difference among the three periods was found in only multiparous women (*p* = 0.021).

Comparisons of game usage, daily life behavior, and thoughts about game usage among Group A, Group B, and Group C for each trimester are shown in Table 2(a–c). In both primiparous women and multiparous women, the proportion of partners who used games was significantly higher in Group A in all pregnancy periods. In early pregnancy, sleeping hours in multiparous women were significantly longer in Group A than in Group C (*p* = 0.029). There were significant differences in the proportions of women who responded that they could not sleep due to game use among Group A, Group B, and Group C in both primiparous women and multiparous women. A significant difference among Group A, Group B, and Group C in the proportions of women who responded that game usage by mothers produces a negative environment for child/children’s development was found only in primiparous women. In mid-pregnancy and late pregnancy, there was a significant difference in the proportions of primiparous women with regularity of wake-up time and bedtime among Group A, Group B, and Group C. For primiparous women in the three gestational periods and multiparous women in late pregnancy, there were significant differences among Group A, Group B, and Group C in the proportions of women who had anxiety that they might have a game addiction. For primiparous women in mid-pregnancy and multiparous women in early and late pregnancy, there were significant differences among Group A, Group B, and Group C in the proportions of women who thought that they could not use gaming sufficiently due to pregnancy and child-rearing.

### 3.4. Internet Gaming Disorder Scale (IGDS)

In the 238 women in early pregnancy, one woman (0.4%) had 5 points for the IGDS, one woman (0.4%) had 4 points, one woman (0.4%) had 3 points, four women (1.7%) had 2 points, 14 women (5.9%) had 1 point, and 218 women (91.6%) had 0 points. In mid-pregnancy, none (0.0%) had more than 4 points, one woman (0.4%) had 3 points, three women (1.3%) had 2 points, 10 women (4.2%) had 1 point, and 224 women (94.1%) had 0 points. In late pregnancy, none (0.0%) had more than 4 points, one woman (0.4%) had 3 points, two women (0.8%) had 2 points, five women (2.1%) had 1 point, and 230 women (96.6%) had 0 points. In the early pregnancy period, there was only one woman with a score of more than 5 points, which indicates a high level of dependence on Internet games. There were no significant differences in the proportion of the IGDS scores between primiparous women and multiparous women in the early, mid, and late pregnancy periods. In this study, Cronbach’s alpha value was 0.58.

## 4. Discussion

This study is the first longitudinal survey on game usage among pregnant women. We showed that approximately half of the pregnant women did not engage in any game use. Since women who did not play games during pregnancy (Group C) included a significantly higher proportion of women who responded that game usage by mothers produces a negative environment for child/children’s development, this may be one of the reasons that women did not engage in any game use.

On the other hand, approximately one-fourth of the pregnant women continued their game usage during three periods of pregnancy (Group A). For women in Group A, there was an increase in game usage time in late pregnancy in primiparous women, but there were no changes in the three periods in multiparous women. In addition, the game usage time in primiparous women was longer than that in multiparous women in both early pregnancy and late pregnancy. We found that primiparous women may use their pre-birth maternity leave in late pregnancy to play games. Multiparous women understand that their physical condition might not be good after childbirth and that they will have less leisure time due to childcare. Therefore, they may adapt their game usage into their daily lives during pregnancy. However, since primiparous women do not have childcare experience, they may think that they can maintain game usage time after childbirth. Therefore, for primiparous women, midwives need to provide information and opportunities to consider their postpartum life in detail during pregnancy or before pregnancy and guide them to adjust game usage with postpartum daily life in mind.

The other approximately one-fourth of the pregnant women changed their game usage during three periods of pregnancy—they played games during one period but not during another (Group B). In Group B, the proportion of women who used games was 74.2% in early pregnancy, but it decreased to 32.3% in mid-pregnancy and 29.0% in late pregnancy. For primiparous women in Group B, there were significant decreases in game usage time from early pregnancy to mid-pregnancy and late pregnancy. On the other hand, for multiparous women in Group B, there was no significant change in game usage time. The reason for this result might be that these women in Group B decided to continue or discontinue game usage according to their own preferences, and they adapted game usage to the physical and psychological changes caused by pregnancy.

We showed that sleeping hours in late pregnancy were shorter than those in early pregnancy in both primiparous women and multiparous women and that multiparous women had shorter sleeping hours than did primiparous women in all periods. These findings are consistent with the results of previous studies [8,10,11]. Suzuki et al. reported that shorter sleeping hours in multiparous women were due to the burden of childcare [8]. In Japan, it has been reported that the time spent on household and childcare duties by women is longer than that by men [18] and that there are many mothers who feel a significant burden of childcare responsibilities [19]. Multiparous women may have a short sleeping time and a short game usage time since their free time is limited due to housework and childcare.

Women who continued to play games in all three periods (Group A) had no changes in sleeping hours during pregnancy, but they included a high proportion of women with irregular sleep habits. That was noticeable in primiparous women. Since it has been reported that additive gaming might be associated with poor quality of sleep [20], game usage in primiparous women who continued to play games might be a reason for irregular sleeping habits. On the other hand, multiparous women might not have had sleep disorders because their game usage time was not long enough to affect their sleeping states.

It has been reported that sleep disorders during a period of one month before birth could be a risk factor for weak labor in primiparous women [21]. It has also been reported that severely disrupted sleep was related to longer labor duration and that women with severely disrupted sleep were 5.2 times more likely to have cesarean section [22]. Therefore, for primiparous women who continue to use games during pregnancy, it is necessary for midwives to assess the impact of game usage on quality of sleep and to advise them to control their game usage.

We found that women who continued to use games during pregnancy (Group A) had feelings of anxiety that they might have a game addiction and a feeling of dissatisfaction that they could not play games sufficiently due to their pregnant state or childcare, suggesting that they had psychological ambivalence.

Since it has been reported that maternal anxiety and stress can have an impact on neural development in the fetus [23], it is necessary to reduce stress in pregnant women. The late pregnancy period, when mothers feel both excitement about the upcoming birth and anxiety about the process of labor, is a period that requires various preparations for childbirth and parenting. It has been reported that learning about childbirth in childbirth preparation classes is beneficial as a preparation for labor and that such preparation leads to better acceptance of childbirth in the postpartum period [24]. It has also been reported that participants in childbirth preparation classes have interests in both childbirth and parenting preparation and that they highly valued classes conducted by midwives that included these contents [25]. As the approach by midwives is effective in preparing for childbirth and parenting, midwives can inform pregnant women about potential physical and psychological issues that might occur due to the continuation of game usage during pregnancy and make pregnant women control their game usage.

In women who continued game usage during pregnancy (Group A), the proportion of partners who used games was high in both primiparous women and multiparous women. It has been reported that women engage in gaming in response to their partner’s gaming behavior since game usage is a shared recreational activity among couples [26]. We previously reported that women whose partners engage in game usage are more likely to continue gaming after pregnancy [7]. For the issues of game use during pregnancy, an approach for not only pregnant women but also their partners is needed.

Midwives should inform both women and their partners about potential issues that could occur from the continuation of game usage during pregnancy in childbirth preparation classes or before pregnancy. It is important for partners to recognize the impact of their own gaming behaviors and control their game usage.

There are several limitations in the present study. As this study utilized an online survey, it is restricted to individuals with internet access, potentially leading to a dominance of specific characteristics. The responses are based on the participants’ subjective experiences. In the present study, we could not show that short game usage time is better for pregnancy progress or perinatal events in pregnant women, although we compared various factors according to gaming time. A prospective comparative study between short and long game usage times is needed in the future. The differences in game usage time between multiparous women and primiparous women in early pregnancy might be due to the time needed for childcare by multiparous women. A question about game usage during the first pregnancy in multiparous women might be needed. In this study, we found that the use of games by pregnant women is associated with the gaming behavior of their partners. Further surveys regarding partners’ game usage are necessary. In this prospective study, 79 women dropped out after mid-pregnancy. There were 50 women who were unable to decide on a birth facility due to the COVID-19 pandemic and were transferred to other hospitals after mid-pregnancy. Out of the distributed QR codes, 29 women did not respond, but we did not ask the reasons for their non-response. Further research is needed to examine the potential issues faced by pregnant women who continue to use games during pregnancy, the gaming habits of partners, and the required educational approaches and support.

## 5. Conclusions

An increase in game usage time was found at late pregnancy in primiparous women who continued to use games during pregnancy. Those primiparous women had psychological ambivalence about their game use, and they had a higher proportion of partners who used games. Midwives need to include partners in their approach and should discuss with couples about game usage and advise them to control game usage in daily life. From a public health perspective, providing information about game usage and establishing a support system to promote appropriate usage and a balanced lifestyle are necessary.

## Figures and Tables

**Figure 1 healthcare-11-03017-f001:**
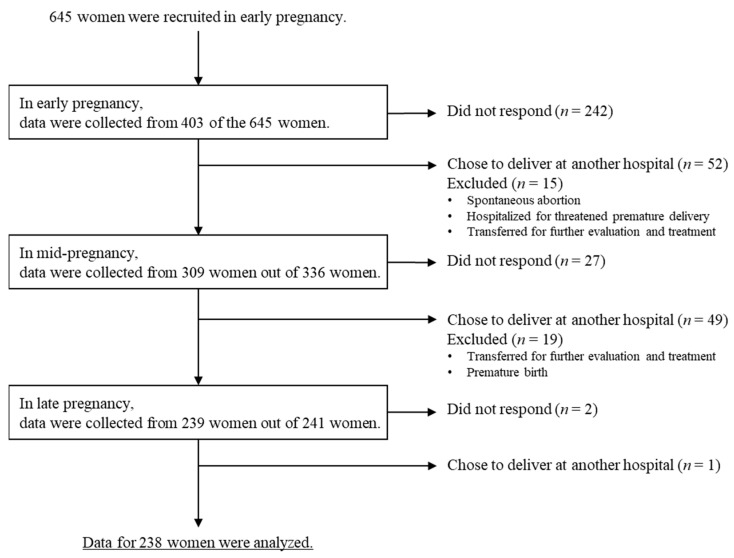
Flow diagram of the prospective study.

**Figure 2 healthcare-11-03017-f002:**
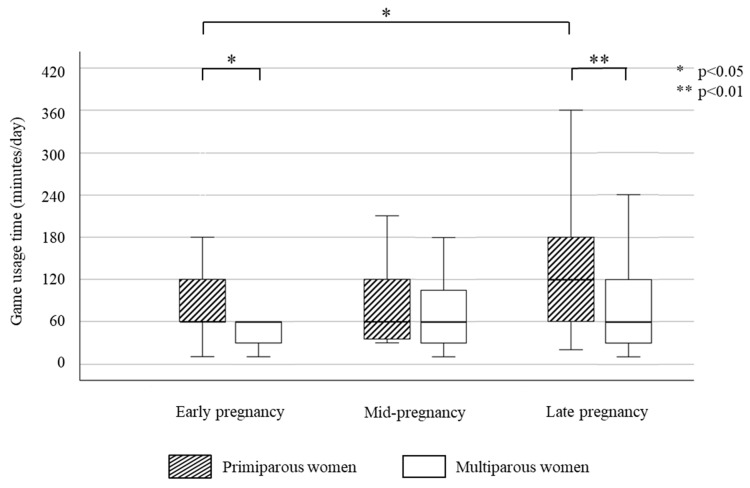
Changes in game usage time of pregnant women who played games during all pregnancy periods (Group A) for primiparous women and multiparous women. Friedman’s test (Primiparous women: *p* = 0.002, Multiparous women: *p* = 0.302).

**Table 1 healthcare-11-03017-t001:** Changes in background characteristics and various factors among the three periods in primiparous women and multiparous women.

	Primiparous (*n* = 113)								Multiparous (*n* = 125)							
	Early Pregnancy ^a^		Mid-Pregnancy ^b^		Late Pregnancy ^c^				Early Pregnancy ^a^		Mid-Pregnancy ^b^		Late Pregnancy ^c^			
	*n*	%	*n*	%	*n*	%	*p* Value	Post Hoc ^d^	*n*	%	*n*	%	*n*	%	*p* Value	Post Hoc ^d^
Game use ^f^																
Yes	55	48.7	42	37.2	41	36.3	0.107		46	36.8	38	30.4	42	33.6	0.563	
No	58	51.3	71	62.8	72	63.7			79	63.2	87	69.6	83	66.4		
Partner’s game use ^f^																
Yes	88	77.9	82	72.6	82	72.6	0.573		84	67.2	80	64.0	79	63.2	0.782	
No	25	22.1	31	27.4	31	27.4			41	32.8	45	36.0	46	36.8		
Sleeping hours (hours/day) ^g^	7.6 ± 1.4		7.2 ± 1.1		7.0 ± 1.5		<0.001	E > L	7.2 ± 1.4		6.8 ± 1.0		6.5 ± 1.2		<0.001	E > L
Regularity of wake-up time and bedtime ^f^																
Yes	91	80.5	99	87.6	87	77.0	0.110		107	85.6	116	92.8	101	80.8	0.021	M > L
No	22	19.5	14	12.4	26	23.0			18	14.4	9	7.2	24	19.2		
Presence of days on which you could not sleep due to game use ^e^																
Yes	4	3.5	3	2.7	1	0.9	0.544		3	2.4	4	3.2	1	0.8	0.545	
No	109	96.5	110	97.3	112	99.1			122	97.6	121	96.8	124	99.2		
Presence of days on which you could not cook your own meal due to game use ^e^																
Yes	4	3.5	3	2.7	0	0.0	0.169		0	0.0	0	0.0	1	0.8	1.000	
No	109	96.5	110	97.3	113	100.0			125	100.0	125	100.0	124	99.2		
Presence of days on which you could not eat regularly (3 meals/day) due to game use ^e^																
Yes	1	0.9	0	0.0	1	0.9	1.000		1	0.8	0	0.0	0	0.0	1.000	
No	112	99.1	113	100.0	112	99.1			124	99.2	125	100.0	125	100.0		
Frequency of eating ready-to-eat meals (e.g., instant food, precooked food and fast food) ^f^																
≥3 days/week	33	29.2	21	18.6	28	24.8	0.173		46	36.8	25	20.0	29	23.2	0.006	E > M
≤2 days/week	80	70.8	92	81.4	85	75.2			79	63.2	100	80.0	96	76.8		
I think that game usage by mothers produces a negative environment for child/children’s development ^f^																
Yes	79	69.9	64	56.6	68	60.2	0.103		80	64.0	78	62.4	75	60.0	0.806	
No	34	30.1	49	43.4	45	39.8			45	36.0	47	37.6	50	40.0		
I think that game usage is addictive ^f^																
Yes	99	87.6	97	85.8	100	88.5	0.830		107	85.6	109	87.2	109	87.2	0.912	
No	14	12.4	16	14.2	13	11.5			18	14.4	16	12.8	16	12.8		
I think that I may have a game addiction ^f^																
Yes	16	14.2	10	8.8	11	9.7	0.390		9	7.2	11	8.8	8	6.4	0.763	
No	97	85.8	103	91.2	102	90.3			116	92.8	114	91.2	117	93.6		
I think that I could not use gaming sufficiently due to pregnancy and child-rearing ^f^																
Yes	7	6.2	5	4.4	3	2.7	0.433		12	9.6	7	5.6	4	3.2	0.103	
No	106	93.8	108	95.6	110	97.3			113	90.4	118	94.4	121	96.8		

^a^ Information was obtained in early pregnancy. ^b^ Information was obtained in mid-pregnancy. ^c^ Information was obtained in late pregnancy. ^d^ E: Early pregnancy, M: Mid-pregnancy, L: Late pregnancy. ^e^ Fisher’s exact test. ^f^ Chi-square test (If there were statistically significant differences, post hoc analysis with the Z test was conducted with a Bonferroni correction). ^g^ Friedman’s test. Post hoc analysis with the Wilcoxon signed-rank test was conducted with a Bonferroni correction. Values of sleeping hours are indicated as means ± standard deviation. *n*: number.

**Table 2 healthcare-11-03017-t002:** Comparison of background characteristics and various factors among the three groups in primiparous women and multiparous women. (**a**) Early pregnancy. (**b**) Mid-pregnancy. (**c**) Late pregnancy.

**(a)**																
	**Primiparous Women (*n* = 113)**								**Multiparous Women (*n* = 125)**							
**Early Pregnancy**	**Group A**		**Group B**		**Group C**				**Group A**		**Group B**		**Group C**			
	** *n* **	**%**	** *n* **	**%**	** *n* **	**%**	***p* Value**	**Post Hoc ^a^**	** *n* **	**%**	** *n* **	**%**	** *n* **	**%**	***p* Value**	**Post Hoc ^a^**
Game use																
Yes	32	100.0	23	74.2	0	0.0			27	100.0	19	59.4	0	0.0		
No	0	0.0	8	25.8	50	100.0			0	0.0	13	40.6	66	100.0		
Partner’s game use ^c^																
Yes	30	93.8	24	77.4	34	68.0	0.023	A > C	23	85.2	24	75.0	37	56.1	0.014	A > C
No	2	6.3	7	22.6	16	32.0			4	14.8	8	25.0	29	43.9		
Sleeping hours (hours/day) ^d^	7.3 ± 1.3		7.9 ± 1.7		7.5 ± 1.2		0.506		7.8 ± 1.9		6.9 ± 1.0		6.7 ± 1.1		0.035	A > C
Regularity of wake-up time and bedtime ^b, c^																
Yes	24	75.0	22	71.0	45	90.0	0.071 ^c^		21	77.8	26	81.3	60	90.9	0.157 ^b^	
No	8	25.0	9	29.0	5	10.0			6	22.2	6	18.8	6	9.1		
Presence of days on which you could not sleep due to game use ^b^																
Yes	4	12.5	0	0.0	0	0.0	0.010	A > C	3	11.1	0	0.0	0	0.0	0.009	A > C
No	28	87.5	31	100.0	50	100.0			24	88.9	32	100.0	66	100.0		
Presence of days on which you could not cook your own meal due to game use ^b^																
Yes	3	9.4	0	0.0	1	2.0	0.202		0	0.0	0	0.0	0	0.0		
No	29	90.6	31	100.0	49	98.0			27	100.0	32	100.0	66	100.0		
Presence of days on which you could not eat regularly (3 meals/day) due to game use ^b^																
Yes	1	3.1	0	0.0	0	0.0	0.558		1	3.7	0	0.0	0	0.0	0.216	
No	31	96.9	31	100.0	50	100.0			26	96.3	32	100.0	66	100.0		
Frequency of eating ready-to-eat meals (e.g., instant food, precooked food and fast food) ^c^																
≥3 days/week	11	34.4	13	41.9	9	18.0	0.053		13	48.1	12	37.5	21	31.8	0.332	
≤2 days/week	21	65.6	18	58.1	41	82.0			14	51.9	20	62.5	45	68.2		
I think that game usage by mothers produces a negative environment for child/children’s development ^c^																
Yes	18	56.3	20	64.5	41	82.0	0.034	A < C	14	51.9	21	65.6	45	68.2	0.322	
No	14	43.8	11	35.5	9	18.0			13	48.1	11	34.4	21	31.8		
I think that game usage is addictive ^b^																
Yes	28	87.5	26	83.9	45	90.0	0.668		21	77.8	29	90.6	57	86.4	0.366	
No	4	12.5	5	16.1	5	10.0			6	22.2	3	9.4	9	13.6		
I think that I may have a game addiction ^b^																
Yes	9	28.1	6	19.4	1	2.0	0.001	A = B > C	4	14.8	3	9.4	2	3.0	0.103	
No	23	71.9	25	80.6	49	98.0			23	85.2	29	90.6	64	97.0		
I think that I could not use gaming sufficiently due to pregnancy and child-rearing ^b^																
Yes	5	15.6	1	3.2	1	2.0	0.046	A = B = C	5	18.5	5	15.6	2	3.0	0.020	A > C
No	27	84.4	30	96.8	49	98.0			22	81.5	27	84.4	64	97.0		
(**b**)																
	**Primiparous Women (*n* = 113)**								**Multiparous Women (*n* = 125)**							
**Mid-Pregnancy**	**Group A**		**Group B**		**Group C**				**Group A**		**Group B**		**Group C**			
	** *n* **	**%**	** *n* **	**%**	** *n* **	**%**	***p* Value**	**Post Hoc ^a^**	** *n* **	**%**	** *n* **	**%**	** *n* **	**%**	***p* Value**	**Post Hoc ^a^**
Game use																
Yes	32	100.0	10	32.3	0	0.0			27	100.0	11	34.4	0	0.0		
No	0	0.0	21	67.7	50	100.0			0	0.0	21	65.6	66	100.0		
Partner’s game use ^c^																
Yes	30	93.8	21	67.7	31	62.0	0.006	A > B = C	23	85.2	21	65.6	36	54.5	0.020	A > C
No	2	6.3	10	32.3	19	38.0			4	14.8	11	34.4	30	45.5		
Sleeping hours (hours/day) ^d^	6.9 ± 0.8		7.5 ± 1.1		7.2 ± 1.2		0.124		7.1 ± 1.2		6.9 ± 1.0		6.7 ± 1.3		0.627	
Regularity of wake-up time and bedtime ^b^																
Yes	24	75.0	27	87.1	48	96.0	0.019	A < C	24	88.9	30	93.8	62	93.9	0.658	
No	8	25.0	4	12.9	2	4.0			3	11.1	2	6.3	4	6.1		
Presence of days on which you could not sleep due to game use ^b^																
Yes	2	6.3	0	0.0	1	2.0	0.459		1	3.7	2	6.3	1	1.5	0.420	
No	30	93.8	31	100.0	49	98.0			26	96.3	30	93.8	65	98.5		
Presence of days on which you could not cook your own meal due to game use ^b^																
Yes	1	3.1	1	3.2	1	2.0	1.000		0	0.0	0	0.0	0	0.0		
No	31	96.9	30	96.8	49	98.0			27	100.0	32	100.0	66	100.0		
Presence of days on which you could not eat regularly (3 meals/day) due to game use ^b^																
Yes	0	0.0	0	0.0	0	0.0			0	0.0	0	0.0	0	0.0		
No	32	100.0	31	100.0	50	100.0			27	100.0	32	100.0	66	100.0		
Frequency of eating ready-to-eat meals (e.g., instant food, precooked food and fast food) ^c^																
≥3 days/week	5	15.6	6	19.4	10	20.0	0.877		5	18.5	10	31.3	10	15.2	0.171	
≤2 days/week	27	84.4	25	80.6	40	80.0			22	81.5	22	68.8	56	84.8		
I think that game usage by mothers produces a negative environment for child/children’s development ^c^																
Yes	17	53.1	14	45.2	33	66.0	0.165		14	51.9	17	53.1	47	71.2	0.098	
No	15	46.9	17	54.8	17	34.0			13	48.1	15	46.9	19	28.8		
I think that game usage is addictive ^b^																
Yes	26	81.3	28	90.3	43	86.0	0.621		20	74.1	30	93.8	59	89.4	0.080	
No	6	18.8	3	9.7	7	14.0			7	25.9	2	6.3	7	10.6		
I think that I may have a game addiction ^b^																
Yes	6	18.8	4	12.9	0	0.0	0.003	A = B > C	3	11.1	3	9.4	5	7.6	0.770	
No	26	81.3	27	87.1	50	100.0			24	88.9	29	90.6	61	92.4		
I think that I could not use gaming sufficiently due to pregnancy and child-rearing ^b^																
Yes	4	12.5	1	3.2	0	0.0	0.018	A > C	4	14.8	1	3.1	2	3.0	0.084	
No	28	87.5	30	96.8	50	100.0			23	85.2	31	96.9	64	97.0		
(**c**)																
	**Primiparous Women (*n* = 113)**								**Multiparous Women (*n* = 125)**							
**Late Pregnancy**	**Group A**		**Group B**		**Group C**				**Group A**		**Group B**		**Group C**			
	** *n* **	**%**	** *n* **	**%**	** *n* **	**%**	***p* Value**	**Post Hoc ^a^**	** *n* **	**%**	** *n* **	**%**	** *n* **	**%**	***p* Value**	**Post Hoc ^a^**
Game use																
Yes	32	100.0	9	29.0	0	0.0			27	100.0	15	46.9	0	0.0		
No	0	0.0	22	71.0	50	100.0			0	0.0	17	53.1	66	100.0		
Partner’s game use ^c^																
Yes	29	90.6	21	67.7	32	64.0	0.024	A > B = C	24	88.9	18	56.3	37	56.1	0.008	A > B = C
No	3	9.4	10	32.3	18	36.0			3	11.1	14	43.8	29	43.9		
Sleeping hours (hours/day) ^d^	6.6 ± 1.5		7.3 ± 1.6		7.0 ± 1.5		0.355		6.9 ± 1.3		6.7 ± 1.1		6.3 ± 1.3		0.201	
Regularity of wake-up time and bedtime ^b, c^																
Yes	20	62.5	23	74.2	44	88.0	0.025 ^c^	A < C	21	77.8	26	81.3	54	81.8	0.912 ^b^	
No	12	37.5	8	25.8	6	12.0			6	22.2	6	18.8	12	18.2		
Presence of days on which you could not sleep due to game use ^b^																
Yes	1	3.1	0	0.0	0	0.0	0.558		0	0.0	1	3.1	0	0.0	0.472	
No	31	96.9	31	100.0	50	100.0			27	100.0	31	96.9	66	100.0		
Presence of days on which you could not cook your own meal due to game use ^b^																
Yes	0	0.0	0	0.0	0	0.0			0	0.0	0	0.0	1	1.5	1.000	
No	32	100.0	31	100.0	50	100.0			27	100.0	32	100.0	65	98.5		
Presence of days on which you could not eat regularly (3 meals/day) due to game use ^b^																
Yes	0	0.0	0	0.0	1	2.0	1.000		0	0.0	0	0.0	0	0.0		
No	32	100.0	31	100.0	49	98.0			27	100.0	32	100.0	66	100.0		
Frequency of eating ready-to-eat meals (e.g., instant food, precooked food and fast food) ^c^																
≥3 days/week	7	21.9	7	22.6	14	28.0	0.777		5	18.5	10	31.3	14	21.2	0.440	
≤2 days/week	25	78.1	24	77.4	36	72.0			22	81.5	22	68.8	52	78.8		
I think that game usage by mothers produces a negative environment for child/children’s development ^c^																
Yes	17	53.1	16	51.6	35	70.0	0.163		12	44.4	17	53.1	46	69.7	0.051	
No	15	46.9	15	48.4	15	30.0			15	55.6	15	46.9	20	30.3		
I think that game usage is addictive ^b^																
Yes	28	87.5	27	87.1	45	90.0	0.869		23	85.2	26	81.3	60	90.9	0.328	
No	4	12.5	4	12.9	5	10.0			4	14.8	6	18.8	6	9.1		
I think that I may have a game addiction ^b^																
Yes	8	25.0	3	9.7	0	0.0	<0.001	A > C	5	18.5	2	6.3	1	1.5	0.008	A > C
No	24	75.0	28	90.3	50	100.0			22	81.5	30	93.8	65	98.5		
I think that I could not use gaming sufficiently due to pregnancy and child-rearing ^b^																
Yes	3	9.4	0	0.0	0	0.0	0.040	A = B = C	3	11.1	1	3.1	0	0.0	0.015	A > C
No	29	90.6	31	100.0	50	100.0			24	88.9	31	96.9	66	100.0		

(**a**) Information was obtained in early pregnancy. *n*: number ^a^ A: Group A, B: Group B, C: Group C ^b^ Fisher’s exact test. ^c^ Chi-square test (If there were statistically significant differences, post hoc analysis with the Z test was conducted with a Bonferroni correction). ^d^ Kruskal–Wallis test (If there were statistically significant differences, post hoc analysis with the Mann–Whitney U test was conducted with a Bonferroni correction). Values of sleeping hours are indicated as means ± standard deviation. (**b**) Information was obtained in mid-pregnancy. *n*: number ^a^ A: Group A, B: Group B, C: Group C ^b^ Fisher’s exact test. ^c^ Chi-square test (If there were statistically significant differences, post hoc analysis with the Z test was conducted with a Bonferroni correction). ^d^ Kruskal–Wallis test. Values of sleeping hours are indicated as means ± standard deviation. (**c**) Information was obtained in late pregnancy. *n*: number ^a^ A: Group A, B: Group B, C: Group C ^b^ Fisher’s exact test. ^c^ Chi-square test (If there were statistically significant differences, post hoc analysis with the Z test was conducted with a Bonferroni correction). ^d^ Kruskal–Wallis test. Values of sleeping hours are indicated as means ± standard deviation.

## Data Availability

The data presented in this study are not publicly available because of privacy restrictions.

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
