# Peer review of "Differences in Changes in Game Usage Time and Game Use-Related Factors Depending on Parity in a Prospective Study of Pregnant Women in Japan"

_healthcare, 2023, doi:10.3390/healthcare11233017_

Round 1

Reviewer 1 Report

Comments and Suggestions for Authors

This is a well written manuscript, Although I did not find anything new.

Comments on the Quality of English Language

this is a well written study.

Author Response

Thank you so much for taking time out of your busy schedule to review our manuscript. Please see the attachment.

Reviewer 2 Report

Comments and Suggestions for Authors

Good afternoon, congratulations on your work. Below, I offer some suggestions and corrections regarding your work.

Abstract

1. Follow the publication guidelines and present a structured abstract. Also, use specific data and do not present the results in abstract terms.

Materials and Methods

2. In this section, propose several subsections such as design, sample, statistical analysis, etc. Propose a structured writing of this section to facilitate understanding for the reader.

3. Which test did you use to analyze the normality of the variables under study and decide between parametric and non-parametric tests? Why do you mix them?

4. Define the IGDS questionnaire as well as its scoring and what it means.

Results

5. Add Cronbach's alpha values and McDonald's omega for the questionnaire used. Is it validated for that population?

Discussion

6. Restructure the discussion in shorter paragraphs to facilitate understanding of the text.

Author Response

(The authors gave the same response as above.)

Reviewer 3 Report

Comments and Suggestions for Authors

Title: Differences in changes in game usage time and game use-related factors depending on parity in a prospective study of 3 pregnant women in Japan

General Concept of Article: Paper wants to examine Game Usage and how it affected pregnant women in a longitudinal study.  The goal was to explore if there were any detrimental effects of excessive game usage by pregnant women throughout their pregnancy and how they perceive game usage during pregnancy.  The comparison between those who experience greater parity or whose partners also play games and how this was associated with game was usage was a prominent sub aim.  The study was done on 645 pregnant women who visited a birth center in Japan where 238 people responded throughout the study period.

General Comments:  The paper concept is interesting as game usage might be influential in women who are pregnant.  My biggest concerns are the following:

(1) The motivation for how game usage in general is related to detrimental outcomes is not well motivated.  This should be treated more thoroughly in the introduction.  Motivated by how literature strongly supports this research.  There were only a couple of papers that touched up on this.  This needs more development.

(2) The description of groups, analysis, study design (where data is collected) needs more detail.  There was not enough detail provided here in methods.

(3) The statistical analysis proposed seems to be incorrect.  The study collects information over time on participants.  This means that there are repeated measures collected on participants.  None of the proposed analysis considers this fact.  For example, when comparing within primiparous and multiparous women at each stage, Chi-Square tests, KW tests were used.  These all assume independence.  The only test that considers repeated measures was Friedman’s test. 

Additionally, the study considered multiple tests across many variables as opposed to simply considering a multivariable model.  It is not clear why this was not done.  If there are 3 outcomes, then adjusting for baseline characteristics would be more appropriate.

A more appropriate modeling procedure would be considering longitudinal mixed models to account for repeated measures.  Additionally, you can then include all women who did respond at each step (403, 309 and 239) at each step.  These models allow for differential response at each step. 

There was no discussion of why those who responded at one stage were excluded as there might be material reasons that they did not respond to follow up at each step.  So using mixed models would assist with this.

Specific Comments by Section

Abstract

You noted 645 were recruited but this is misleading given your final sample size; additionally, no formal statistics/results are provided.  Given the abstract is often used to screen out/in literature it would be good to provide more results for the reader here to capture the findings.

Intro: The overall discussion is reasonable, but more support is required here to contextualize why game usage is a problem.  Covering more of the literature about potential detrimental effects of game usage to highlight why this is important would be helpful.  As it stands, the review seems sparse.  Additionally, is it game usage or excessive game usage that is the issue?  If it is excessive usage, I would characterize it that way.  Simply using game doesn’t seem to be an issue.  However, excessive or addictive game usage seems like that is what the authors are trying to capture.

Methods: For readers who are not familiar with birth centers, provide more insight (line 62-69); can you provide location and how they were selected?  Is it the case that 645 of 693 people agreed to participate?

What was the reason for excluding unmarried participants (line 87)?  To examine parity excluding women with multiple pregnancies seems like a strange decision as this variable would be required to understand its association with outcomes of interest.

-The flow diagram was helpful.  See comments below however about statistical methods.

Line 127 – how did you determine the game use groups?  There needs to be much greater discussion here as these are the main groups that you are exploring differences here but no context about how you defined these is provided.

For statistical methods you have a longitudinal study, but your methods don’t appear to reflect that.  Did you do a longitudinal analysis to account for the repeated measurements?  If not, then the problem with the analysis would be that independence is violated and the analysis is not correct.  You cannot examine how proportions, average, medians, etc are different if there are repeated measures at each time point and the methods used assume independence.  This is not correct.

General thoughts:  Break up the paper into parts here – have study design and data collection; data description/instruments used; statistical methods.  I would have these as subheadings for each of the methods. You did this for the results, and it helped with readability but didn’t do this for methods.

Additionally a major methodological error seems to be made.

Results

As noted, you performed Chi-Square/ANOVA and KW tests across these groups, but these assume independence of measures.  Since you collected on information on women who persisted to the study there is a repeated measure.  This analysis is not correct.  Someone who responded in each group contributes to the value at each the statistic created at each time point.  You cannot do this.  This is like doing an independent samples t-test on someone who is in both groups.  The appropriate test would be a paired t-test in that case.  So for something like this, you need to consider a repeated measures design.  If you didn’t collect information on each person where you can identify them at each step, then you can only compare ACROSS the groups, but making a comparison within the groups as noted in Table 1, 2A, 2b, etc. is not correct.

Discussion: Not discussed as the analysis is not performed correctly.

Author Response

(The authors gave the same response as above.)

Reviewer 4 Report

Comments and Suggestions for Authors

Materials and Methods section

It would be better to add subheadings to make the test clearer: like sample size and study participants, questionnaires and variables, and statistical analysis.

line 62, be more precise about the location of the birth center!

For the background variables can you give us a further explanation, like putting in brackets categories of employment, smoking, and alcohol habits?

Results, The description of the results may be shortened!

3.1. Background characteristics, lines 140-149. The authors named the first section of questions "Background characteristics" and then under 3.1. subsection they are writing about the second and third sections of questions (game usage time and sleeping time). Please, describe in two sentences the main background characteristics of the sample like occupational status, smoking and drinking habits.

Results, 3.3. The authors should explain groups in the method section and here just to present results.

Results 3.4. Instead of telling us how much each woman had points for the IGDS just translate this into the level of dependence on Internet games like high, medium, and low for example.

Discussion, maybe to add as a limitation the use of online questionnaires and subjective responses of the participants!

The conclusion should be wider in terms of the public health approach and not only related to midwives and their counseling. What are the policy implications of the game usage problem?

Comments on the Quality of English Language

Minor editing of English language required.

Author Response

(The authors gave the same response as above.)

Reviewer 5 Report

Comments and Suggestions for Authors

The authors conducted a longitudinal survey that delves into the game usage habits of pregnant women. There are some areas that might benefit from further clarification and enhancement:

- While the method used is clearly outlined, it would be helpful to delve deeper into the reasons for the dropouts, particularly the 29 women who did not respond to the distributed QR codes. This might offer a clearer perspective on the sample being studied.

- In various sections of the discussion, there are passages that might benefit from clearer structuring and brevity to enhance readability and information absorption.

- I suggest exploring further the gaming habits of multiparous women during their first pregnancy. This could provide additional insights into the observed differences between primiparous and multiparous women.

- Your study highlights the influence of partners on the gaming habits of pregnant women. I suggest emphasizing this aspect further, possibly considering a separate survey on the gaming habits of partners to better understand their perspectives.

- In your conclusions, I recommend emphasizing more the importance of an inclusive approach that encompasses both pregnant women and their partners. This might provide a more comprehensive view of potential solutions and guidelines to be adopted.

- Finally, since this is the inaugural study of its kind, it would be beneficial to provide a section on potential future research, suggesting possible areas of inquiry and open questions.

Comments on the Quality of English Language

The text should be revised by a native speaker to remove several typos.

Author Response

(The authors gave the same response as above.)

Round 2

Reviewer 5 Report

Comments and Suggestions for Authors

The authors have made the changes requested in the previous review.

Comments on the Quality of English Language

Minor editing of English language required